# RGB-D-Based Stair Detection and Estimation Using Deep Learning

**DOI:** 10.3390/s23042175

**Published:** 2023-02-15

**Authors:** Chen Wang, Zhongcai Pei, Shuang Qiu, Zhiyong Tang

**Affiliations:** School of Automation Science and Electrical Engineering, Beihang University, Beijing 100191, China

**Keywords:** stair detection, RGB-D, deep learning, multimodality, stair geometric parameter estimation

## Abstract

Stairs are common vertical traffic structures in buildings, and stair detection tasks are important in environmental perception for autonomous mobile robots. Most existing algorithms have difficulty combining the visual information from binocular sensors effectively and ensuring reliable detection at night and in the case of extremely fuzzy visual clues. To solve these problems, we propose a stair detection network with red-green-blue (RGB) and depth inputs. Specifically, we design a selective module, which can make the network learn the complementary relationship between the RGB feature maps and the depth feature maps and fuse the features effectively in different scenes. In addition, we propose several postprocessing algorithms, including a stair line clustering algorithm and a coordinate transformation algorithm, to obtain the stair geometric parameters. Experiments show that our method has better performance than existing the state-of-the-art deep learning method, and the accuracy, recall, and runtime are improved by 5.64%, 7.97%, and 3.81 ms, respectively. The improved indexes show the effectiveness of the multimodal inputs and the selective module. The estimation values of stair geometric parameters have root mean square errors within 15 mm when ascending stairs and 25 mm when descending stairs. Our method also has extremely fast detection speed, which can meet the requirements of most real-time applications.

## 1. Introduction

In the environmental perception research of autonomous mobile robots, stair climbing is a fundamental problem and its premise is the detection and estimation of stairs. For stair detection, there are two kinds of methods according to the method of stair feature extraction: line extraction methods and plane extraction methods.

Line extraction methods abstract the features of stairs as a collection of continuous lines in an image and use relevant algorithms to extract the features of lines in RGB or depth maps. There are two methods for feature extraction: traditional image processing and deep learning. The former use Canny edge detection, a Hough transform, and other algorithms to extract lines in images or use local surface orientations to extract lines in point clouds [1,2,3]. The latter use deep learning computer vision to make a network extract line features through learning on a dataset with annotations [4]. The plane extraction methods regard the stairs as a group of continuously distributed parallel planes in point clouds. Point clouds are divided into planes using plane segmentation algorithms, and then the stair planes can be extracted according to some manually designed rules [5,6]. Additionally, some methods first use deep learning to locate a region of interest (ROI) containing stairs and then use traditional image processing algorithms for line extraction within the ROI [7,8]. These methods often have poor real-time performance because of the two detection stages.

According to the information input mode for a detection method, stair detection can also be divided into monocular and binocular detection methods. Monocular detection methods extract features only from single-mode perceptual information. Some common workflows are as follows. Line extraction methods are applied to obtain stair features in RGB maps, and the corresponding key points are obtained from the point clouds. Then, the stair geometric parameters are obtained [9]. Line extraction methods can also be applied to obtain stair features in depth maps to obtain the stair geometric parameters [10]. Plane segmentation methods are applied to obtain the stair planes in point clouds to calculate the stair geometric parameters [5,6]. The sensors used in monocular detection methods may be binocular, and only the single-mode information is used for feature extraction. Binocular detection methods extract features from multimodal perceptual information. Some common workflows are as follows. After using line extraction methods to extract features from RGB and depth maps, the features from different modalities are fused artificially using certain rules [11,12]. Monocular detection methods often have better real-time performance but lower accuracy than binocular detection methods.

The above methods solve the problems of stair detection and recognition for autonomous robots used in buildings to some extent. However, these methods have difficulty ensuring reliable stair detection in some challenging scenes. For line extraction methods, the edges of stairs in an RGB map are clear when ascending stairs but fuzzy when descending stairs, which is the opposite in a depth map, namely, the RGB and depth map information are complementary to some extent. Therefore, monocular detection methods alone cannot obtain complete and reliable visual clues. For binocular detection methods, the detection results from the RGB and depth maps need to be fused. Existing fusion methods are mainly based on artificially designed rules, which often do not have wide applicability. For plane extraction methods, algorithms that directly use plane segmentation algorithms to process the point clouds returned from a depth sensor often have poor real-time performance, which is caused by the very large amount of point cloud data.

In recent years, deep learning technology has been applied not only in image classification but also in various fields of computer vision [13]. In previous works, among methods based on line extraction, the deep learning method StairNet [4] has achieved breakthrough accuracy and extremely fast speed under the condition of monocular vision by virtue of its novel stair feature representation method and end-to-end convolution neural network (CNN) architecture. However, due to the limitations of monocular vision, the method has poor performance in night scenes with extreme lighting, scenes with extremely fuzzy visual clues, and scenes with a large number of objects similar to stairs. To overcome the shortcomings of StairNet, we find that the RGB image and depth image are complementary to some extent and add depth map input into the network on the basis of StairNet. Figure 1 shows some RGB images and the corresponding depth images on the dataset.

We can see that the depth images have clear and sharp edges when descending stairs, but the edges in the RGB images are often fuzzy. The opposite is true when ascending stairs. This is caused by the generation principle of the depth images; that is, the depth images have clear and sharp edges where the distance changes suddenly. In addition, depth images are clearer than RGB images at night. To explore the complementary relationship between the RGB feature maps and the depth feature maps, we propose a selective module, which makes the network combine the information from both modalities in different scenes effectively.

In summary, the highlights of this research are mainly reflected in three aspects: (1) Based on StairNet, we add depth map input into the network, and through a selective module, the network can learn the complementary relationship between the RGB feature maps and the depth feature maps so that a feature fusion process can be integrated into the neural network and eliminate the artificially designed rules. We call this network StairNetV2. Compared with previous methods, we can obtain extremely fast detection speed while processing multimodal perception information and still have excellent performance in extreme lighting scenes at night, scenes with extremely fuzzy visual cues, and scenes with a large number of objects similar to stairs. (2) We propose several postprocessing algorithms to obtain the stair geometric parameters, including a stair line clustering algorithm based on the least squares method and a coordinate transformation algorithm based on attitude angles. (3) We provide an RGB-D dataset with fine annotations. There are 2388 RGB images and the corresponding 2388 depth images in the training set and 608 RGB images and the corresponding 608 depth images in the validation set. The stair line classification and the locations of the two endpoints are stored in each label file.

The remainder of the paper is organized as follows. Section 2 introduces some RGB-D-based deep learning methods and some fusion methods for deep learning. Section 3 describes the methodology of the proposed method that includes the following main components: (1) the architecture of StairNetV2 with multimodal inputs and the selective module; (2) the loss function with dynamic weights; (3) the stair line clustering algorithm based on the least squares method; (4) the estimation of stair geometric parameters. Section 4 shows the experimental results and evaluates the effectiveness of our method. Section 5 discusses the results. Section 6 concludes the paper.

## 2. Related Works

Most RGB-D-based stair detection methods use traditional image processing algorithms for feature extraction, and due to the limitation of image processing algorithms, the combination methods of RGB and depth features are often rigid [1,11,12]. Most deep learning stair detection methods [4,7,8] focus on extracting stair features in monocular vision through a CNN, and there is no deep learning method to make full use of the complementary relationship between the RGB map and the depth map for stair detection. Regarding the RGB-D fusion methods for deep learning, some methods fuse features in the input and output locations by simple summation and concatenation [14,15,16,17,18], and some methods design special modules to explore the implicit relationship between the two modalities [19,20,21,22]. This section briefly introduces some RGB-D-based stair detection methods and some RGB-D fusion methods for deep learning.

### 2.1. RGB-D-Based Stair Detection Methods

Stair detection methods based on image processing algorithms always combine RGB and depth features through artificially designed rules. In Ref. [11], the RGB and depth maps are obtained using a depth camera, and then prior knowledge is used to determine which map is used for detection. Finally, stair lines are obtained through edge and line detection algorithms. In Ref. [12], the RGB and depth maps are processed at the same time, and the edge information in the maps is obtained. Then, the local binary pattern (LBP) feature is extracted from the RGB map, and the one-dimensional depth feature is extracted from the depth map. Finally, the features are combined and classified using a support vector machine (SVM). Ref. [1] first applies a Sobel operator to extract the edges from the RGB map, and then a Hough transform is applied to extract the straight lines from the edge image. Finally, a one-dimensional depth feature is extracted from the depth image to distinguish stairs from objects with textures similar to stairs, such as pedestrian crosswalks. Ref. [14] develops an RGB-D stair recognition system that applies unsupervised domain adaptation learning to help visually impaired people navigate independently in unfamiliar environments. In this study, the three-channel RGB input is adjusted to a four-channel RGB-D input to fuse depth information.

### 2.2. RGB-D Fusion Methods for Deep Learning

With the development of low-cost depth sensors, there are increasingly more RGB-D-based deep learning methods in computer vision. However, there is no established methodology to perfectly fuse these two modalities inside a CNN [23]. Some common fusion methods are as follows. Refs. [15,16] directly adjust the three-channel RGB input to the four-channel RGB-D input, and the depth information is sent to the network as the fourth input channel. Refs. [17,18] design two parallel branches to process RGB images and depth images and then concatenate the feature vectors obtained from each branch to fuse the features. Finally, the number of channels is adjusted, and the softmax function is applied to obtain the final classification result. This simple way to fuse RGB and depth information in the input or output cannot make full use of the implicit relationship between the RGB map and the depth map. To solve this problem, Ref. [19] proposes canonical correlation analysis (CCA) to fuse the two kinds of features. Ref. [24] applies ensemble learning to combine an RGB map model and a depth map model. Ref. [20] proposes a multimodal layer to combine the information from the RGB stream and the depth stream. Through multimodal learning, this layer can find the discriminative features of each modality and harness the complementary relationship between them. Ref. [21] first trains an RGB network and a depth network separately and then deletes their softmax classification layers after training. Finally, a fusion network with the softmax function is trained to obtain the final classification. Ref. [22] applies an attention feature complementary (AFC) module for RGB-D fusion to make the informative features obtain higher values of weights, and the complementary information from depth maps can be exploited more effectively. Ref. [25] proposes a depth-similar convolution operation, which converts the unique pixel relationship in the depth map into depth-similar weight dots with the corresponding convolution kernel.

## 3. Methods

Our method mainly involves four aspects, including the architecture of StairNetV2 with the selective module, the loss function with dynamic weights, stair line clustering based on the least squares method, and the method used to obtain the stair geometric parameters based on attitude angles.

### 3.1. Network Architecture

As described above, we add depth map input to the network based on StairNet and fuse the features by a selective module. On this basis, to accelerate postprocessing of the network output results, we adjust the size of the network output to 32 × 16. To make the classification result more robust, we adjust the previous 3 classifications to 2 classifications; that is, we no longer judge whether a stair line is a concave line or a convex line but only judge whether each cell contains stair lines. This means that we only distinguish between the foreground and background, and the judgment of concave and convex lines can be easily obtained through prior knowledge. In summary, as shown in Figure 2, a full-color image and a depth image with a size of 512 × 512 are sent into the network and processed with a fully convolutional architecture. The backbone contains five downsampling operations, and we obtain feature maps with a size of 32 × 16 after the final downsampling. The output has two branches: a classification branch with a size of 32 × 16 × 1 output tensor and a location branch with a size of 32 × 16 × 8 output tensor. The output tensor obtained from the classification branch is used to predict the probability that a cell contains stair lines, and the output tensor obtained from the location branch regresses two sets of normalized values to deal with cells containing two stair lines. The left endpoint locations are represented using (x1, y1) and (x3, y3), and the right endpoint locations are represented using (x2, y2) and (x4, y4) in each cell.

Compared with the StairNet architecture, the receptive field is further expanded due to the increase in downsampling times, so atrous spatial pyramid pooling (ASPP) [26] is removed. When going up and down stairs, the stair lines tend to be horizontally distributed in the vision field. StairNet applies asymmetric hole convolution to take advantage of this prior knowledge. In StairNetV2, we take advantage of this prior knowledge by designing the loss function, and the asymmetric hole convolution in the squeeze-and-excitation (SE)-ResNeXt [27,28] block is replaced with the standard hole convolution. In addition, we retain the focus module [29] as the initial downsampling operation, and the depth branch also has a focus module. Table 1 shows the detailed network architecture of StairNetV2.

Different from some existing methods of fusing RGB feature maps and depth feature maps through simple summation and concatenation, we apply the softmax function to explore the complementary relationship between the RGB stream and the depth stream. First, the feature maps containing RGB information and the feature maps containing depth information are added element by element. Then, global average pooling is applied to obtain a 1 × 1 × c vector as the feature descriptor of the depth or RGB feature maps, and the softmax activation function is applied to the channel c dimension to obtain two feature descriptors that restrict each other. Finally, the feature descriptor is multiplied by the corresponding original feature maps element by element, and the depth and RGB feature maps are added to obtain the mixed feature maps. Figure 3 shows the structure of the selective module.

### 3.2. Loss Function with Dynamic Weights

The loss function of StairNet consists of classification loss and location loss, as shown in Equation (Equation 1):(1)$$\begin{array}{c} L\left(\left\{p_{ij}\right\},\left\{x_{ij}\right\},\left\{y_{ij}\right\}\right)=\frac{1}{MN}\sum_{i}^{M}\sum_{j}^{N}\left(L_{cls}\left(p_{ij},p_{ij}^{*}\right)+\lambda \left(p_{ij}L_{loc}\left(x_{ij},x_{ij}^{*}\right)+\alpha p_{ij}L_{loc}\left(y_{ij},y_{ij}^{*}\right)\right)\right) \end{array}$$
where $L_{cls}$ and $L_{loc}$ represent classification loss and location loss, respectively, and $L_{loc}$ is divided into abscissa loss and ordinate loss. *M* and *N* represent the height and width of the output tensors, namely, 32 × 16. The indexes *i* and *j* represent the pixel coordinates in the width direction and height direction, respectively. $p_{ij}$ and $p_{ij}^{*}$ are one-dimensional vectors that represent the probability of containing stair lines and the corresponding ground truth, respectively. $x_{ij}$ and $x_{ij}^{*}$ are four-dimensional vectors that represent the four predicted abscissa coordinates and corresponding ground truth, respectively. $y_{ij}$ and $y_{ij}^{*}$ are four-dimensional vectors that represent the four predicted ordinate coordinates and corresponding ground truth, respectively. λ and α are the weight coefficients, λ is used to allocate the weight between the classification loss and the location loss, and α is used to allocate the weight between the abscissa location loss and the ordinate location loss. In StairNet, λ and α are set to 4.

This method of setting the weights to fixed values provides a direction for the learning and convergence of the network to a certain extent. However, the values of weights should be different at different stages of network convergence. Therefore, we propose a loss function with dynamic weights, and Equation (Equation 1) is rewritten as:(2)$$\begin{array}{c} L\left(\left\{p_{ij}\right\},\left\{x_{ij}\right\},\left\{y_{ij}\right\}\right)=\frac{1}{MN}\sum_{i}^{M}\sum_{j}^{N}\left(L_{cls}\left(p_{ij},p_{ij}^{*}\right)+\alpha p_{ij}L_{loc}\left(x_{ij},x_{ij}^{*}\right)+\beta p_{ij}L_{loc}\left(y_{ij},y_{ij}^{*}\right)\right) \end{array}$$
where α and β represent the weights of the abscissa location loss and the ordinate location loss, respectively. The remaining parameters are the same as in Equation (Equation 1). At the beginning of the training, α and β are set to 10. After each epoch, α and β are adjusted dynamically according to the evaluation results on the validation set. The evaluation method investigates the total error of the abscissa and ordinate prediction location of the positive class on the validation set, which are recorded as $X_{error}$ and $Y_{error}$, respectively. The corresponding formulas are as follows:(3)$$\begin{array}{c} X_{error}=\sum_{k=0}^{V}\sum_{i}^{M}\sum_{j}^{N}p_{ij}^{*}E_{loc}\left(x_{ij},x_{ij}^{*}\right) \end{array}$$
(4)$$\begin{array}{c} Y_{error}=\sum_{k=0}^{V}\sum_{i}^{M}\sum_{j}^{N}p_{ij}^{*}E_{loc}\left(y_{ij},y_{ij}^{*}\right) \end{array}$$
where *V* represents the number of samples on the validation set, and *i*, *j*, *M*, *N*, $p_{ij}^{*}$, $x_{ij}$, $x_{ij}^{*}$, $y_{ij}$ and $y_{ij}^{*}$ have the same meanings as in Equation (Equation 1). $E_{loc}$ represents the error evaluation method, and the L1-norm is used here. After obtaining $X_{error}$ and $Y_{error}$, we use Equations (Equation 5) and (Equation 6) to adjust α and β. It should be noted that if the adjusted α and β are less than threshold σ (set as 0.5 in the study), they will not be adjusted.
(5)$$\begin{array}{c} \alpha_{i}=\alpha_{i-1}+\frac{X_{error}-Y_{error}}{Max(X_{error},Y_{error})} \end{array}$$
(6)$$\begin{array}{c} \beta_{i}=\beta_{i-1}-\frac{X_{error}-Y_{error}}{Max(X_{error},Y_{error})} \end{array}$$
where *i* represents the current epoch, *i* − 1 represents the previous epoch, and *i*≥ 1. $Max(X_{error},Y_{error})$ represents the larger of $X_{error}$ and $Y_{error}$.

We apply the loss function with dynamic weights to enable the network to adjust its focus on the horizontal and vertical coordinate location loss. By designing a reasonable loss function, we use the prior knowledge about the distribution feature of stair lines in an image. Compared with the method in StairNet that uses prior knowledge by designing a module with a special structure, our method has better flexibility and wider applicability.

### 3.3. Stair Line Clustering

For methods based on line extraction, the broken lines need to be connected before obtaining the final stair line. Ref. [30] proposes an edge linking and tracking algorithm to eliminate the small breaks or gaps for the long horizontal edges. Refs. [10,12] apply the brute-force algorithm [31] to link short edges and obtain a long edge. However, these algorithms can only deal with horizontal stair lines and have high time complexity, which are not applicable for our method.

As described above, through StairNetV2, the stair structured information we finally obtained is stored in two tensors, and because of the diversity of shooting angles, most stair lines are at a certain angle with the horizontal direction. To quickly connect sloping stair lines in each cell, we propose a stair line clustering algorithm based on the least squares method. Inspired by Ref. [32], we combine the adjacent information from the classification output tensor and the geometry information from the location output tensor to connect the lines efficiently. To make full use of the network outputs, we need to cluster the cells belonging to the same stair lines. As shown in Figure 4, the steps of the stair line clustering algorithm are as follows. First, the initial slope k and intercept b are calculated using the least squares method according to the coordinates in the cells of the middle two columns, namely, column 7 and column 8. Then, the remaining cells are clustered from the middle to both sides to determine which line each cell covers. If they are on existing lines, they are added to the corresponding set of lines. Otherwise, new lines are added to the results. Finally, k and b are recalculated using the least squares method, and the process is repeated until clustering is completed.

We select the middle two columns as the initial set of lines because the camera is often facing the stairs in the actual scene, and there are more stair lines in the middle area. The prediction results are more accurate because of more visual information than in the side area. In addition, when the predicted (x1, y1, x2, y2) and (x3, y3, x4, y4) are very close, we only take the former.

### 3.4. Estimation of Stair Geometric Parameters

In Section 3.3, we sort the output results of the neural network through stair line clustering based on the least squares method and finally obtain the point sets and the corresponding slope and intercept describing each stair line. Since the slope and intercept reflect the information of all point sets, we use the six-tuple data (x1, y1, x2, y2, k, b) to describe a stair line, where x1 and y1 represent the left endpoint, x2 and y2 represent the right endpoint, and k and b represent the slope and intercept, respectively. To make full use of the location information of the fitted stair line, we take 4 sampling points at symmetrical and equal intervals on the left and right sides of the image middle axis so that we can obtain 9 sampling points. Next, we apply the aligned point cloud data to obtain the camera coordinates of the 9 sampling points, as shown in Figure 5.

After obtaining the camera coordinates of 9 sampling points, we fit them to obtain a final set of three-dimensional coordinates to calculate the stair geometric parameters. We apply the least squares method of lines in three-dimensional space [33] to fit these 9 sampling points, and the spatial straight line equation can be expressed as the intersecting line of two planes, as shown in Equation (Equation 7):(7)$$\left\{\begin{array}{c} x=k_{1}z+b_{1}\\ y=k_{2}z+b_{2} \end{array}\right.$$
where, $k_{1}$, $k_{2}$, $b_{1}$ and $b_{2}$ are the parameters to be determined. *x*, *y*, and *z* are the three-dimensional coordinates in space. After the parameters are determined, let *x* = 0 to obtain $z=-\frac{b_{1}}{k_{1}}$ and $y=\frac{k_{1}b_{2}-k_{2}b_{1}}{k_{1}}$. Therefore, we can obtain the intersection point of the stair line and the camera YOZ plane. However, the intersection points are obtained in the camera coordinate system, and we need to transfer these points to the world coordinate system to calculate the stair geometric parameters, which requires the camera’s attitude for coordinate transformation. In our research, Intel’s Realsense D435i depth camera [34] is used, and its inertial measurement unit (IMU) module can obtain the three components gx, gy, and gz of gravity acceleration. In the world coordinate system defined by the D435i depth camera, the axis is vertically downward, the $Z_{c}$ axis is horizontally forward and the $X_{c}$, $Y_{c}$ and $Z_{c}$ axes meet the right-hand rule, as shown in Figure 6a. We use the attitude angles to describe the attitude of the camera. The attitude angles include pitch, roll, and yaw. The pitch angle is the angle between the camera $Z_{c}$ axis and XOZ plane of the world coordinate system, and the head up direction of the camera is the positive direction. The roll angle is the angle between the camera $Y_{c}$ axis and the vertical plane containing the camera $Z_{c}$ axis, and the right tilt direction of the camera is the positive direction. The yaw angle is always 0 because the Z axis of the world coordinate system and the $Z_{c}$ axis of the camera are in the same vertical plane. The pitch angle and roll angle can be calculated using Equation (Equation 8).
(8)$$\left\{\begin{array}{l} \mathrm{roll}=arcsin\left(\frac{-\mathrm{gx}}{\sqrt{{\mathrm{gx}}^{2}+{\mathrm{gy}}^{2}+{\mathrm{gz}}^{2}}}\right)\\ \mathrm{pitch}=arcsin\left(\frac{\mathrm{gz}}{\sqrt{{\mathrm{gx}}^{2}+{\mathrm{gy}}^{2}+{\mathrm{gz}}^{2}}}\right) \end{array}\right.$$

The above world coordinate system is defined by the IMU of the D435i depth camera, and its Z-axis points to the rear of the camera. For the convenience of calculation, as shown in Figure 6b, we redefine the world coordinate system $X_{w}Y_{w}Z_{w}$ by rotating the original world coordinate system 180° around the Y-axis, namely, it has a reverse X-axis and Z-axis compared with the original world coordinate system. After obtaining the pitch angle and roll angle, the camera coordinates (xc, yc, zc) can be transferred to the world coordinates (xw, yw, zw) through the coordinate transformation formula. Finally, we need to transfer the world coordinates (xw, yw, zw) to the stair coordinates (xs, ys, zs). According to the architectural design principles, the $Y_{s}$ axis and $X_{s}{\mathrm{OZ}}_{s}$ plane of the stair coordinate system should coincide with the world coordinate system. Namely, only the angle between the $Z_{s}$ axis and the $Z_{w}$ axis is needed. This angle can also be understood as the yaw angle of the camera relative to the stair, as shown in Figure 6c. We use two sampling points on the same stair line to obtain the yaw angle.

After obtaining the coordinates (xs, ys, zs) in the stair coordinate system, we can obtain the cross section of the stair, as shown in Figure 6d. We can distinguish whether to go up or down the stairs using very simple prior knowledge; that is, when going up the stairs, we can see convex and concave lines at the same time, but when going down the stairs, we only see convex lines. Whether going up or down the stairs, we use Equation (Equation 9) to calculate the width and height of the stairs:(9)$$\left\{\begin{array}{l} \mathrm{height}=|{\mathrm{ys}}_{i+1}-{\mathrm{ys}}_{i}|\\ \mathrm{width}=|{\mathrm{zs}}_{i+1}-{\mathrm{zs}}_{i}| \end{array}\right.$$
where, *i* represents the *i*-th stair line detected, and the value range is [0, the total number of detected stair lines −1]. *i* +1 represents the stair line closest to the *i*-th stair line on the side away from the camera. For the case of the ascending direction, since concave lines and convex lines can be detected at the same time, the calculated height values or width values of the stair lines on the same horizontal or vertical plane will be close to zero. Therefore, we filter the calculation results using Equation (Equation 9). We discard values less than a threshold value ω and ω = 0.05 m in our study.

## 4. Experimental Results

This section describes several conducted experiments in detail, including the experimental settings, ablation experiments for the selective module and the loss function with dynamic weights, performance experiments for the performance of models under daytime and nighttime scenes, stair line clustering experiments, comparison experiments with existing methods, and stair geometric parameter estimation experiments.

### 4.1. Experimental Settings

#### 4.1.1. Dataset Introduction

We use a Realsense D435i depth camera to capture RGB images and aligned depth images in several actual scenes. These images are segmented, padded, and scaled to 512 × 512. The RGB-D dataset has a total of 5992 images, including 2996 RGB images and the corresponding 2996 depth images. We select 2388 RGB images and corresponding depth images as the training set and 608 RGB images and corresponding depth images as the validation set by random selection.

We use the five-tuple data (cls, x1, y1, x2, y2) to represent a stair line. Although StairNetV2 does not classify stair lines, to ensure the completeness of annotation information, we still label stair lines according to two classifications represented by cls. cls has two values of 0 and 1, representing the convex line and concave line, respectively; x1, y1 and x2, y2 represent the left and right endpoints in the pixel coordinate system, respectively.

#### 4.1.2. Training Strategy

As mentioned above, our network takes an RGB image and the corresponding depth image with a size of 512 × 512 as inputs and the whole training process is conducted for 200 epochs with a batch size of 4. During the training, the Adam optimizer is applied, the initial learning rate is set to 0.001, and the learning rate is halved every 50 epochs. Regarding the loss function, we apply the loss function with dynamic weights introduced in Section 3.2. After each epoch, the weights are dynamically adjusted according to the evaluation. The models are trained on a workstation platform that has an R9 5950X CPU and an RTXA4000 GPU, and the PyTorch framework is applied to implement our method. Considering the uneven distribution of stairs in the dataset, we apply a random flip in the horizontal direction with 0.5 probability as a means of data enhancement.

#### 4.1.3. Evaluation Metrics

We basically follow the evaluation method of StairNet, using the accuracy, recall and intersection over union (IOU) [35] when the confidence is 0.5 as the evaluation index. The confidence calculation is the same as that of StairNet. The difference is that StairNetV2 does not classify stair lines, and we do not focus on the background class. Therefore, the frequency weighted intersection over union (FWIOU) [35] of StairNet can be simplified to only calculate the IOU of the class containing stair lines. The formulas of accuracy, recall, and IOU are as follows: (10)$$\left\{\begin{array}{l} \mathrm{accuracy}=\frac{TP}{TP+FP}\\ \mathrm{recall}=\frac{TP}{TP+FN}\\ \mathrm{IOU}=\frac{TP}{TP+FP+FN} \end{array}\right.$$
where *TP*, *FP*, and *FN* represent the number of true positives, false positives, and false negatives, respectively, when the confidence is 0.5. Specifically, *TP* represents the number of cells predicted by the network to include stair lines, which actually include stair lines. *FP* represents the number of cells predicted by the network to include stair lines but do not actually include stair lines. *FN* represents the number of cells predicted by the network to contain no stair lines but actually contain stair lines.

For the evaluation of the stair geometric parameter estimation, considering factors such as the moving speed of the equipment, the sparsity of the remote point cloud, and the measurement error of the depth camera, we only evaluate the geometric parameters of the three stair steps closest to the camera. We use the absolute error and relative error between the estimation results and the actual geometric sizes of the stairs for evaluation. In addition, to show the adaptability of the algorithm in different scenes, we select some stairs with different building structures and textures for evaluation.

### 4.2. Ablation Experiments

We conduct several ablation studies to demonstrate the effectiveness of the selective module and the loss function with dynamic weights. Since the selective module belongs to the category of network architecture and the loss function with dynamic weights belongs to the category of network training process, we study them separately. First, we try to put the selective module into different network positions, which are after the second downsampling, after the third downsampling, and after the fourth downsampling. The results are shown in Table 2.

It can be seen in Table 2 that when the selective module is located after the second downsampling, the network has the best performance, which indicates that it is unreasonable to simply place a structure for feature fusion such as the selective module in a certain position of the network. For StairNetV2, the network can obtain better performance by extracting the features of the two branches in the shallow layers (bottleneck1.0–1.2) and then further extracting the features of the fused information in the deep layers (bottleneck2.0–3.7).

After the selective module position is determined, we use the loss function with fixed weights and the loss function with dynamic weights to conduct a loss function ablation experiment. Table 3 shows the ablation experiment results of the loss function.

It can be seen that using the loss function with dynamic weights to train the network can improve the network performance, which shows that the reasonable adjustments of some hyperparameters in the network training process can not only accelerate network convergence but also improve model performance.

### 4.3. Performance Experiments

In the performance experiments, we test our models on the validation set, including the model size, inference speed, accuracy, recall, and IOU. Similar to StairNet, there are three versions of StairNetV2, including StairNetV2 1×, StairNetV2 0.5×, and StairNetV2 0.25×. We use a channel width factor to multiply the number of channels to adjust the parameter quantity of the model. The three models are tested on a typical desktop platform and a typical mobile platform. Table 4 shows the specific platforms, parameter quantities of the models, and inference speeds.

It can be seen that the three versions of StairNetV2 all have good real-time performance on both desktop and mobile platforms. Compared with the three models of StairNet, our model is approximately 1/3 the size of the corresponding StairNet model, so we can obtain faster speed. To evaluate the performance of the models more objectively, the validation set is divided into a daytime dataset and a nighttime dataset according to the actual collection scenes, and the accuracy, recall, and IOU of the models are tested on these two datasets. Table 5 shows the accuracy, recall, and IOU of the models on the daytime and nighttime datasets.

The results show that the performance of StairNetV2 1× is the best among the three versions and StairNetV2 0.5× is slightly worse than StairNetV2 1×, while StairNetV2 0.25× is much worse than StairNetV2 0.5× and StairNetV2 1×. The 1× model is more suitable for applications with high accuracy requirements, while the 0.25× model is more suitable for applications with high real-time requirements, and the 0.5× model provides a compromise option. The particularly striking result is the difference in performance between the day and night scenes. The performance of the models is quite close in the day and night scenes, which is significantly improved compared with the performance of StairNet in the night scenes, and this proves the validity of the depth map for the night scenes. There are some visualization detection results including some challenging scenes on the validation set, as shown in Figure 7. The results show that our method still obtains good results in scenes with extreme lighting, different shooting angles, and different stair textures, especially scenes with extremely fuzzy visual cues at night.

### 4.4. Comparison Experiments

In the comparison experiments, the three versions of StairNetV2 are compared with some existing methods, including the deep learning StairNet method, a method based on the traditional image processing algorithms and a method combining deep learning and traditional image processing. For StairNet, because the sizes of the output tensors are different from those of StairNetV2, it is necessary to adjust them to be consistent for comparison. For the traditional image processing method, we use the Gabor filter, Canny edge detection, and Hough transform to implement it. For the method combining deep learning and traditional image processing, a YOLOv5 [29] model is trained to locate the boxes containing stair lines and the traditional image processing algorithms are applied to extract stair lines within the boxes.

Before the experiments, StairNet and StairNetV2 are trained on our training set according to the settings in Section 4.1, and the parameters of traditional image processing algorithms are adjusted to fit our validation set. The results of traditional image processing algorithms are converted to the same format as the results of StairNetV2. We conduct these experiments on the validation set with the evaluation metrics and experimental platforms in Section 4.1. Table 6 shows the detailed performance of the above methods.

The results show that compared with existing state-of-the-art deep learning method, StairNetV2 has a significant improvement in both accuracy and recall, especially the 0.25 × model of StairNetV2, which still significantly surpasses the 1× model of StairNet. Compared with the 0.25× model of StairNet, the 0.25× model of StairNetV2 has faster speed. We benefit greatly from the input of multimodal information and the loss function with dynamic weights. In addition, compared with the methods based on traditional image processing algorithms, deep learning methods benefit from powerful learning ability and still show far more performance than traditional methods. Figure 8 shows some visualization results of the above methods on our validation set. We can see that the deep learning methods have much better performance than the traditional methods in various detection scenes, and StairNetV2 has better performance than StairNet in night scenes and scenes with fuzzy visual clues.

The method based on traditional image processing algorithms needs fine parameter adjustment for Gabor filter, Canny edge detection, and Hough transform, and a set of appropriate parameters are often only applicable to some certain scenes, which leads to poor adaptability of the algorithms. As shown in Figure 8, when the scenes contain a large number of lines that are not stair lines, such as the floor tile gaps in columns 2 and 5 of the subfigures, the algorithms cannot classify the stair lines. The method combining deep learning and traditional image processing can eliminate a large amount of interference information for the traditional algorithms by locating the ROI containing stairs, which greatly reduces the false detection rate, but the improvement of the missing detection problem is ordinary. In addition, when the localization of the object detection algorithm is wrong, the impact on the traditional algorithms will be fatal. For example, as shown in Figure 8b, the object detection algorithm in columns 2 and 3 of the subfigures does not detect the ROI containing stairs, which leads to missing detection of the whole image. StairNet solves the problems of traditional algorithms to a large extent by virtue of its novel stair feature representation method and end-to-end CNN architecture. However, due to the limitations of monocular vision, the method has poor performance in night scenes with extreme lighting, scenes with extremely fuzzy visual clues, and scenes with a large number of objects similar to stairs, as shown in column 2 and columns 4–6 of Figure 8c. Based on StairNet, StairNetV2 adds the depth input and explores the complementary relationship between the RGB image and the depth image by a selective module, which solves the above problems, as shown in Figure 8d.

### 4.5. Stair Line Clustering Experiments

In this section, we test the stair line clustering algorithm. The method, based on the brute-force algorithm, requires that the long edges cover the short edges, which is not applicable to the structured outputs of StarNetV2, so we only test the edge linking and tracking algorithm based on adjacent points as a comparison. We adjust the adjacent point judgment principle of the edge linking and tracking algorithm to fit the sloping stair lines, and we test the inference speeds on the platforms described in Table 4. The results are shown in Table 7.

The results show that compared with the edge linking and tracking algorithm, the stair line clustering algorithm has better real-time performance. For the time complexity of the algorithms, we suppose that there are Q stair lines detected and P cells containing lines, the cells to be clustered are matched with at most Q stair lines in each step of clustering, then the clustering needs at least PQ times for matching, and its time complexity is O(PQ). The algorithm based on adjacent points in Ref. [30] needs to match every two lines, so the time complexity is O(P(P−1)/2), namely, O($P^{2}$). Since Q is far less than P, our algorithm has lower time complexity. Combined with the experimental results, it can be seen that our algorithm has better real-time performance.

### 4.6. Stair Geometric Parameter Estimation Experiments

In this section, we measure the stairs in several actual scenes. We select several detection scenes with different building materials, different building structures, and different lighting conditions for the experiments, as shown in Figure 9. We use the absolute and relative errors for the evaluation of the estimation results and separately evaluate the two walking directions of ascending and descending. Table 8 shows the experimental results of the stair geometric parameter estimation.

The results show that for the case of ascending stairs, the estimations of stair width are accurate, the estimation errors of height are relatively large, and the measured height values are less than the true values in most cases. For the case of descending stairs, the estimation values are not stable. In most cases, the estimation values exceed the true values to some extent. The following are the main error sources of the whole workflow: (1) The error caused by the jitter of the detection algorithm; (2) The error of point cloud data obtained by the depth camera, including the error of distance measurement, the black hole of point cloud at the edges of the stairs, and at the reflective areas of smooth stairs; (3) The error of the IMU leads to the error of the obtained angles, which affects the coordinate transformation.

## 5. Discussion

In this paper, we propose an RGB-D-based deep learning stair detection and estimation method, including StairNetV2 CNN architecture with multimodal inputs and the selective module, the loss function with dynamic weights, the stair line clustering algorithm based on the least squares method and the coordinate transformation algorithm based on attitude angles. First, to solve the problems of StarNet’s poor performance in some challenging scenes, especially night scenes, we add the depth input based on the architecture of StairNet and design the selective module to explore the complementary relationship between the RGB image and the depth image. Next, according to the prior knowledge about the distribution feature of stair lines in an image, we propose the loss function with dynamic weights to optimize the training process. Then, we propose several postprocessing algorithms to estimate the stair geometric parameters, including the stair line clustering algorithm and the coordinate transformation algorithm. The stair line clustering algorithm can quickly process the output tensors of StairNetV2, and the coordinate transformation algorithm can make full use of the location information to obtain the width and height of stairs. Finally, we conduct several experiments to evaluate the proposed method, including ablation experiments for the selective module and the loss function with dynamic weights, performance experiments for StairNetV2, comparison experiments for some existing methods, and the stair geometric parameter estimation experiments for some actual scenes. The experimental results show the effectiveness of the proposed method, and our StairNetV2 has better performance than the existing methods in various detection scenes.

In the history of stair detection research, there have been some RGB-D-based stair detection methods [1,11,12] and some deep learning methods [4,7,8]. However, most RGB-D-based stair detection methods combine the RGB and depth features through artificially designed rules due to the limitation of image processing algorithms, and most deep learning methods extract stair features in monocular vision. The existing RGB-D-based deep learning stair detection methods, as in Ref. [14], can only judge whether the scene contains stairs, which cannot be applied for the environmental perception of autonomous robots. To implement the location of stair lines at the pixel level, most deep learning methods, as in Ref. [8], locate the boxes containing stair lines, and the traditional image processing algorithms are applied to extract stair lines within the boxes, which have poor real-time performance due to the two detection stages.

To completely eliminate the limitations of traditional image processing algorithms, StairNet [4] implements an end-to-end stair line pixel-level location method with deep learning by virtue of its novel stair feature representation method and CNN architecture. However, StairNet has poor performance in night scenes with extreme lighting, scenes with extremely fuzzy visual clues, and scenes with a large number of objects similar to stairs due to the limitation of monocular vision. To overcome the shortcomings of StairNet, we design a novel CNN architecture called StairNetV2 with multimodal inputs and the selective module, and we build an RGB-D stair dataset with depth maps for the training of StairNetV2. The selective module can explore the complementary relationship between the two modalities effectively, so StairNetV2 can realize stable detection in various complex scenes.

StairNetV2 provides a high-performance method for stair detection in images. However, StairNetV2 can still be improved further. The estimation of stair geometric parameters relies on several postprocessing algorithms, which cannot be integrated into CNN. The reason is that our workflow extracts features from images and transfers them to point clouds. In our future work, we will extract the plane features of stairs directly in point clouds using deep learning and build an end-to-end CNN architecture for stair estimation.

## 6. Conclusions

We propose a neural network architecture with multimodal inputs for stair detection and implement a complete RGB-D-based deep learning stair estimation method with several postprocessing algorithms. In this way, we overcome the problem of reliable detection at night and in the case of extremely fuzzy visual clues. We also propose a loss function with dynamic weights that can optimize the network training process. In addition, for RGB-D-based stair detection research, we provide a stair dataset with depth maps and fine annotations. The experimental results demonstrate the performance and effectiveness of the proposed method. The stair geometric parameters have root mean square errors within 15 mm when ascending stairs and 25 mm when descending stairs, which can meet the requirements of most autonomous mobile robots for the stair estimation.

## Figures and Tables

**Figure 1 sensors-23-02175-f001:**
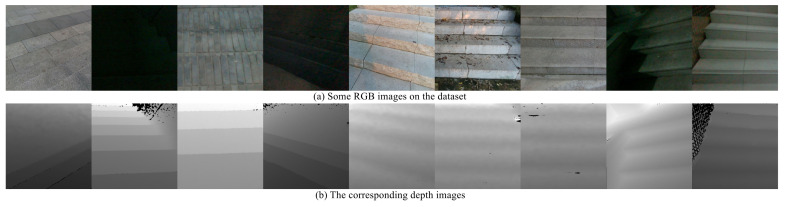
Some RGB images and the corresponding depth images on the dataset. The RGB image and the corresponding depth image are complementary to some extent.

**Figure 2 sensors-23-02175-f002:**
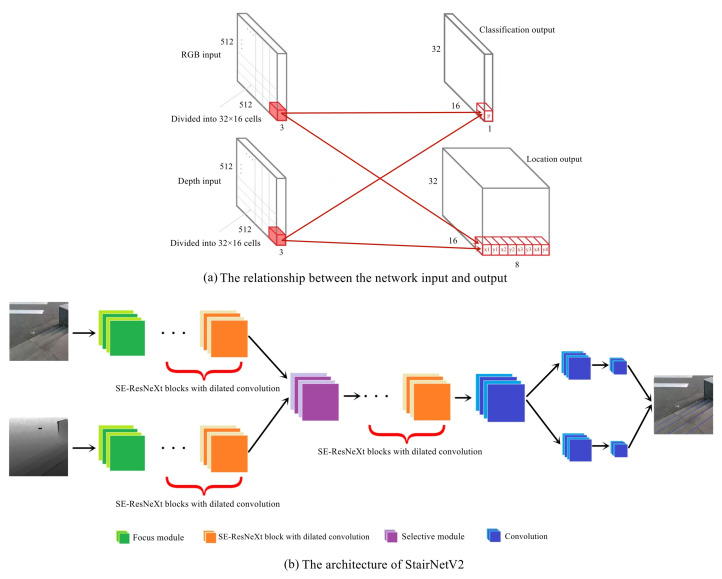
The network architecture of StairNetV2. (**a**) shows that the network has RGB and depth inputs and classification and location outputs. After five downsampling operations, the input size 512 × 512 is transferred to the output size 32 × 16. (**b**) shows the architecture of StairNetV2.

**Figure 3 sensors-23-02175-f003:**
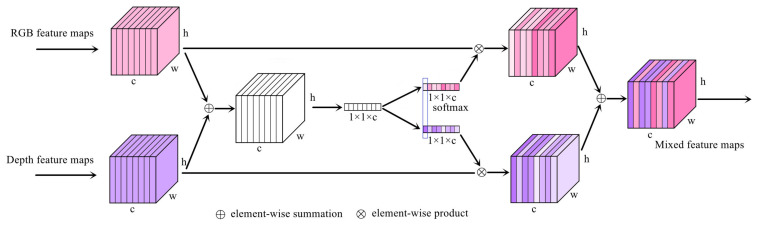
The selective module. The softmax function is applied to explore the complementary relationship between the RGB feature maps and the depth feature maps instead of simple summation and concatenation.

**Figure 4 sensors-23-02175-f004:**
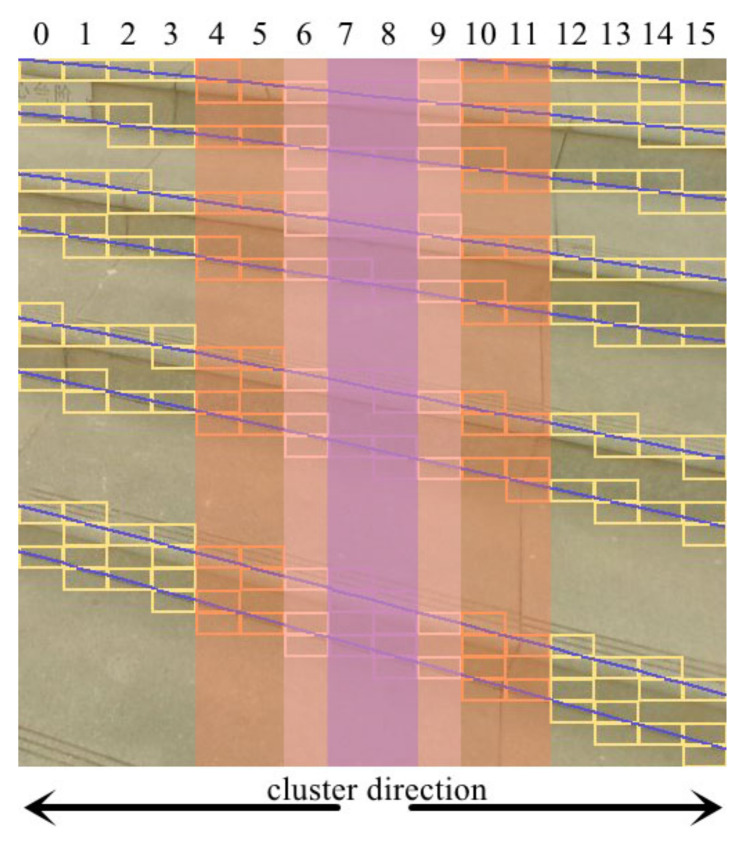
Illustration of stair line clustering based on the least squares method. Purple, pink, orange, and yellow represent the relevant areas of each cluster.

**Figure 5 sensors-23-02175-f005:**
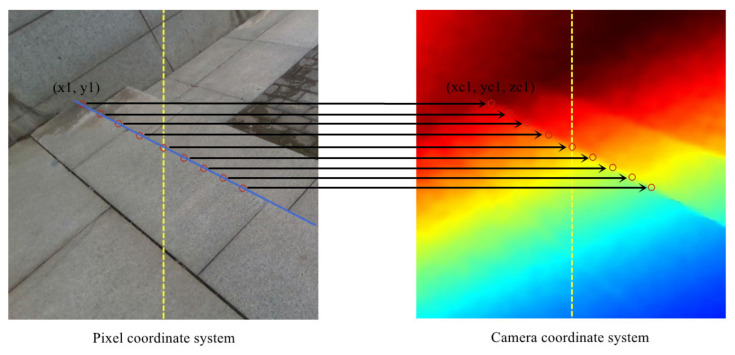
The transformation of stair sampling points. Taking the left endpoint of the stair line as an example, its coordinates in the pixel coordinate system are (x1, y1). Using the aligned point cloud data, its corresponding camera coordinates (xc1, yc1, zc1) are obtained, and the remaining sampling points are the same.

**Figure 6 sensors-23-02175-f006:**
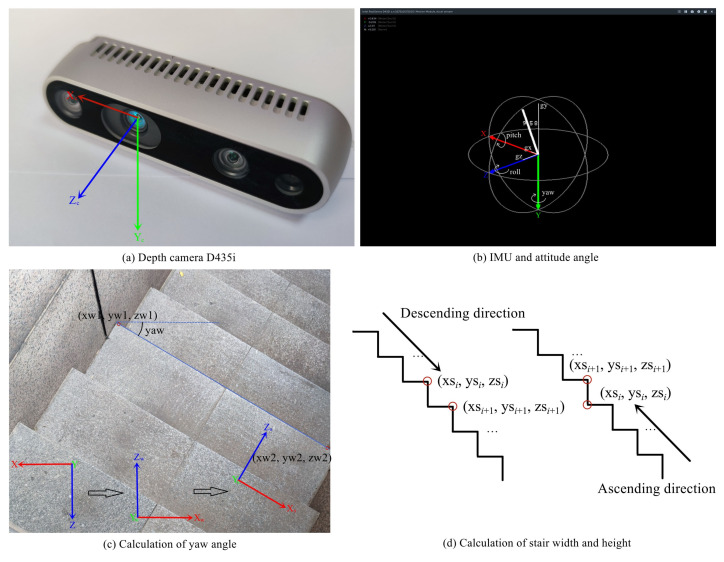
Calculation process of stair width and height. (**a**) shows the Realsense D435i depth camera and the definition of the camera coordinate system. (**b**) shows the definition of the world coordinate system in the IMU module of the D435i depth camera. (**c**) shows the calculation of the yaw angle, which can be calculated through any two points on a stair line in the world coordinate system. (**d**) shows the method of calculating the width and height of stairs in the stair coordinate system.

**Figure 7 sensors-23-02175-f007:**
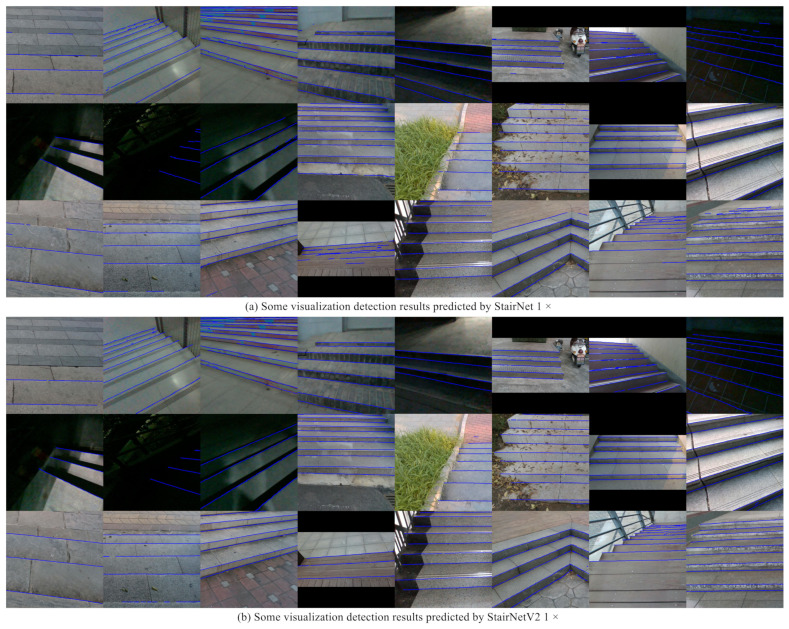
Some visualization detection results predicted by StairNet 1× and StairNetV2 1×.

**Figure 8 sensors-23-02175-f008:**
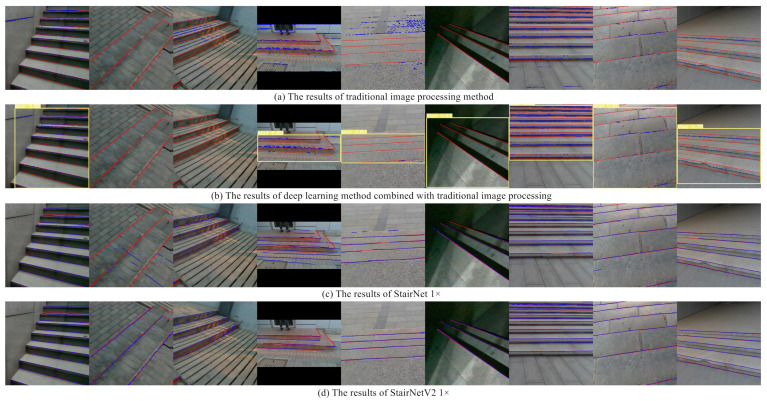
Visualization of the comparison experiments. The red line shows the ground truth and the blue line shows the predicted result.

**Figure 9 sensors-23-02175-f009:**
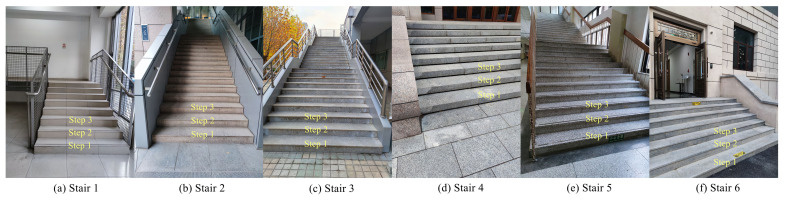
Experimental scenes of the stair geometric parameter estimation experiments.

**Table 1 sensors-23-02175-t001:** StairNetV2 architecture with an input size of 512 × 512. For the output of the location branch, we add a sigmoid activation function to limit the values to (0,1) so that we can obtain the normalized locations of stair lines.

Name		Type		Output Size	
RGB branch	Depth branch	RGB branch	Depth branch	RGB branch	Depth branch
Focus module	Focus module	Tensor slice	Tensor slice	256 × 256 × 32	256 × 256 × 32
Bottleneck 1.0	Bottleneck 1.0	Downsampling	Downsampling	128 × 128 × 128	128 × 128 × 128
Bottleneck 1.1	Bottleneck 1.1			128 × 128 × 128	128 × 128 × 128
Bottleneck 1.2	Bottleneck 1.2			128 × 128 × 128	128 × 128 × 128
Selective module				128 × 128 × 128	
Bottleneck 2.0		Downsampling		64 × 64 × 256	
Bottleneck 2.1		Dilated 2		64 × 64 × 256	
Bottleneck 2.2				64 × 64 × 256	
Bottleneck 2.3		Dilated 4		64 × 64 × 256	
Bottleneck 2.4				64 × 64 × 256	
Bottleneck 2.5		Dilated 8		64 × 64 × 256	
Bottleneck 2.6				64 × 64 × 256	
Bottleneck 2.7		Dilated 16		64 × 64 × 256	
Repeat bottlenecks 2.0 to 2.7				32 × 32 × 256	
Conv 3 × 3		Down sampling with stride = (1,2)		32 × 16 × 128	
Classification	Location	Classification	Location	Classification	Location
Conv 3 × 3	Conv 3 × 3			32 × 16 × 128	32 × 16 × 128
Conv 1 × 1	Conv 1 × 1			32 × 16 × 1	32 × 16 × 8
	Sigmoid		Activation	32 × 16 × 1	32 × 16 × 8

**Table 2 sensors-23-02175-t002:** Ablation experiment results of the selective module.

The Position of the Selective Module	Accuracy (%)	Recall (%)	IOU (%)
After the second downsampling	91.99	93.15	86.16
After the third downsampling	91.52	92.96	85.59
After the fourth downsampling	90.35	92.37	84.08

**Table 3 sensors-23-02175-t003:** Ablation experiment results of the loss function.

Backbone	Loss Function	Accuracy (%)	Recall (%)	IOU (%)
StairNetV2	Fixed weights	91.79	93.15	85.98
StairNetV2	Dynamic weights	91.99	93.15	86.16

**Table 4 sensors-23-02175-t004:** Platforms, parameter quantities of the models, and inference speeds.

Platform	StairNetV2 1 × (10.4 MB)	StairNetV2 0.5 × (2.87 MB)	StairNetV2 0.25 × (0.95 MB)
R9 5950X + RTXA4000	11.09 ms	5.31 ms	3.06 ms
i7 10750H + RTX2060Max-P	19.57 ms	9.62 ms	4.90 ms

**Table 5 sensors-23-02175-t005:** Accuracy, recall, and IOU of the models on the daytime and nighttime datasets.

Model	Accuracy(%)		Recall(%)		IOU(%)	
	Daytime	Nighttime	Daytime	Nighttime	Daytime	Nighttime
StairNetV2 1×	92.88	90.78	93.34	92.87	87.11	84.87
StairNetV2 0.5×	92.05	89.22	93.94	92.91	86.89	83.53
StairNetV2 0.25×	91.06	88.21	92.85	91.87	85.09	81.82

**Table 6 sensors-23-02175-t006:** Comparison experiment results.

Method	Accuracy (%)	Recall (%)	IOU (%)	Runtime (ms)
Gabor + Canny + Hough (our implementation)	26.02	25.20	14.68	7.42
YOLOv5 + Gabor + Canny + Hough (our implementation)	38.56	30.65	20.59	25.17
StairNet 0.25×	83.68	85.59	73.35	3.71
StairNet 0.5×	85.56	85.04	74.37	7.24
StairNet 1×	86.35	85.18	75.07	14.90
StairNetV2 0.25×—Ours	89.85	92.44	83.70	3.06
StairNetV2 0.5×—Ours	90.86	93.51	85.46	5.31
StairNetV2 1×—Ours	91.99	93.15	86.16	11.09

**Table 7 sensors-23-02175-t007:** Experimental results of the stair line clustering algorithm.

Platform	Edge Linking and Tracking [30] (Our Implementation)	Stair Line Clustering
R9 5950X + RTXA4000	16.27 ms	5.72 ms
i7 10750H + RTX2060Max-P	23.13 ms	8.81 ms

**Table 8 sensors-23-02175-t008:** Results of the stair geometric parameter estimation experiments. A cell is denoted as (width, height).

Detection Scenes		True Size (mm)	Estimation Size (mm)		Absolute Error (mm)		Relative Error (%)	
			Ascending	Descending	Ascending	Descending	Ascending	Descending
Stair1	Step1	(296, 160)	(296, 149)	(336, 173)	(0, −11)	(40, 13)	(0.00, −6.88)	(13.51, 8.12)
	Step2	(293, 150)	(282, 141)	(313, 158)	(−11, −9)	(20, 8)	(−3.75, −6.00)	(6.83, 5.33)
	Step3	(295, 150)	(277, 139)	(327, 170)	(−18, −11)	(32, 20)	(−6.10, −7.33)	(10.85,13.33)
Stair2	Step1	(288, 153)	(290, 143)	(302, 156)	(2, −10)	(14, 3)	(0.69, −6.54)	(4.86, 1.96)
	Step2	(289, 162)	(285, 144)	(314, 181)	(−4, −18)	(25, 19)	(−1.38, −11.11)	(8.65, 11.73)
	Step3	(289, 161)	(282, 141)	(321, 160)	(−7,−20)	(32, −1)	(−2.42, −12.42)	(11.07, −0.62)
Stair3	Step1	(304, 153)	(304, 137)	(315, 155)	(0, −16)	(11, 2)	(0.00, −10.46)	(3.62, 1.31)
	Step2	(307, 145)	(304, 134)	(329, 163)	(−3, −11)	(22, 18)	(−0.98, −7.59)	(7.17, 12.41)
	Step3	(307, 145)	(302, 129)	(328, 157)	(−5, −16)	(21, 12)	(−1.63, −11.03)	(6.84, 8.28)
Stair4	Step1	(341, 143)	(346, 126)	(372, 152)	(5, −17)	(31, 9)	(1.47, −11.89)	(9.09, 6.29)
	Step2	(340, 147)	(342, 127)	(372, 145)	(2, −20)	(32, −2)	(0.59, −13.61)	(9.41, −1.36)
	Step3	(342, 146)	(340, 131)	(367, 154)	(−2, −15)	(25, 8)	(−0.58, −10.27)	(7.31, 5.48)
Stair5	Step1	(300, 163)	(301, 150)	(321, 159)	(1, −13)	(21, −4)	(0.33, −7.98)	(7.00, −2.45)
	Step2	(298, 154)	(303, 139)	(313, 162)	(5, −15)	(15, 8)	(1.68, −9.74)	(5.03, 5.19)
	Step3	(296, 154)	(288, 144)	(322, 175)	(−8, −10)	(26, 21)	(−2.70, −6.49)	(8.78, 13.64)
Stair6	Step1	(321, 114)	(321, 108)	(336, 127)	(0, −6)	(15, 13)	(0.00, −5.26)	(4.67, 11.40)
	Step2	(322, 151)	(326, 137)	(332, 148)	(4, −14)	(10, −3)	(1.24, −9.27)	(3.11, −1.99)
	Step3	(322, 152)	(327, 137)	(352, 156)	(5, −15)	(30, 4)	(1.55, −9.87)	(9.32, 2.63)
Root mean square					(6.29, 14.23)	(24.82, 11.40)	(2.11, 9.40)	(8.07, 7.71)

## Data Availability

The dataset presented in this study is openly available at Mendeley Data at https://data.mendeley.com/datasets/p28ncjnvgk (accessed on 9 January 2023), Version 2.

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
