# Peer review of "RGB-D-Based Stair Detection and Estimation Using Deep Learning"

_sensors, 2023, doi:10.3390/s23042175_

Round 1

Reviewer 1 Report

Overall, this paper appears to be good. I have the following concerns:

1, One major contribution of this paper is that " To make full use of the network outputs, we propose a stair line clustering algorithm based on the least squares method,".

However, there are many line clustering methods [1-3]. Is the proposed algorithm best for this application? Experiments should be conducted to compare them.

Z.Z. Wang, "Robust three-dimensional face reconstruction by one-shot structured light line pattern, " Optics and Lasers in Engineering, 124, 105798, 2020. 

2, In figure 7, only the detection results of the proposed method are shown. The detection results of state of the art methods should also be shown for comparison. 

3, English should be improved. 

Reviewer 2 Report

The paper is well-organized and well-structured and also lacks some details for readers to understand. I would suggest the authors address the following concerns in order to meet the publication requirement:

1. Authors are suggested to include more discussion on the results and also include some explanation regarding the justification to support why the proposed method is better in comparison to other methods.

2. Quality of figures is so important too. Please provide some high-resolution figures. Some figures have poor resolution.

3. Does this kind of study have never been attempted before? Justify this statement and give an appropriate explanation to do so in this paper.

4. Provide a more visual comparison between the suggested method and other baselines. The following papers are good examples:

https://doi.org/10.1007/s10479-022-04755-8

https://doi.org/10.1007/978-3-031-04435-9_39

https://doi.org/10.1007/s40747-022-00694-w

Reviewer 3 Report

The topic is good and related to the journal. 

Some comments are given below and the authors should enhance the paper accordingly.

In the abstract section, the authors should talk about the results in more details and the main reason behind getting these results. Also the authors mentioned that Our method also has extremely fast detection speed", How you measure the fast detection here?

I finish the paper and I did not get the meaning or refereing of "RGB"?

In the introduction section, add the strcutre of this paper

No any add form the related work section; at least should add a summazy or anoverview for the selected papers

some related works are worthy to be metioned in this paper like the follwing papers

Salak Image Classification Method Based Deep Learning Technique Using Two Transfer Learning Models

Deep learning based stair detection and statistical image filtering for autonomous stair climbing

Deep leaning-based ultra-fast stair detection

Round 2

Reviewer 1 Report

I would like to recommend this paper for publication in Sensors Journal.

Reviewer 2 Report

The authors addressed all of my concerns and I suggest publishing it.